

# Effectiveness of non-lethal predator deterrents to reduce livestock losses to leopard attacks within a multiple-use landscape of the Himalayan region

Dipanjan Naha, Pooja Chaudhary, Gaurav Sonker and Sambandam Sathyakumar

Department of Endangered Species Management, Wildlife Institute of India, Dehradun, Uttarakhand, India

## ABSTRACT

Lethal measures are widely adopted by local communities and governments to manage human-wildlife conflicts. Such measures lead to large scale decline of carnivore populations globally with trophic cascades on ecosystems and questionable impacts on human-wildlife conflicts. Mitigating human-carnivore conflicts through non-lethal measures will protect endangered predators and secure livelihoods. However, information on the effectiveness of such measures are extremely limited and hence cannot be applied in developing scientific evidence. Further to develop human-carnivore coexistence models, it is important for local community members, biologists and wildlife managers to actively participate in conservation programs. We evaluated the response of a non-lethal visual deterrent (i.e. fox lights) to deter leopard attacks on livestock within a multiple-use landscape of western Himalaya through community engagement. We monitored 16 experimental sites and 17 control sites within 27 villages and recorded data on livestock depredation by leopards between April 2018 to April 2019. A multivariate analysis was conducted to determine the influence of landscape predictors and animal husbandry practices on livestock depredation by leopards within the vicinity of human settlements. We found that visual deterrents discouraged common leopards to predate on livestock (cows and goats). We also demonstrated that community based conservation initiatives are successful in mitigating human-carnivore conflicts within large semi-natural landscapes. We suggest developing site specific coexistence strategies and adopting non-lethal measures to safeguard carnivores, livestock and humans within shared landscapes.

# INTRODUCTION

Large carnivores are apex predators and help regulate the structure and functioning of ecosystems. Decline in populations of apex predators have resulted in degradation of ecological systems, loss of biodiversity and ecosystem services globally (*Ripple et al., 2014*). Loss of wild prey and anthropogenic impacts that degrade and fragment natural ecosystems force large carnivores to share space and resources with humans within larger

Corresponding author
Sambandam
Sathyakumar, ssk@wii.gov.in

heterogeneous landscapes (*Chapron et al., 2014*). As a consequence, large carnivores kill livestock and occasionally attack humans. Economic incentives from wildlife tourism benefit government, private agencies but local community members often share the disproportionate costs of coexistence with large carnivores through livestock losses (*Dickman, 2010*). Financial losses due to livestock predation by large carnivores leads to retaliation and persecution by humans (*Woodroffe, 2000*; *Loveridge et al., 2010*). Livestock depredation is thus regarded as a key stimuli of human-carnivore conflicts globally (*Inskip & Zimmermann, 2009*). Frequent and persistent negative interactions generate antagonism against large carnivores through real or perceived impacts on human wellbeing, safety and livelihoods (*Kansky & Knight, 2014*). Local community members resort to retaliatory killings through poisoning of livestock carcass, bush meat, snaring, spearing, electrocution and shooting of large carnivores (*Inskip et al., 2016*; *Hazzah et al., 2017*). Human-carnivore conflicts also impact the overall ecosystem such as scavengers who die after consuming poisoned meat (*Ogada, 2014*). Hence, effective mitigation measures are urgently required to ensure conservation of large carnivores and functioning of healthy ecosystems.

Lethal control has been widely adopted as the ultimate mitigation strategy to manage human-carnivore conflicts and has been implemented both legally (*Chapron et al., 2014*) and illegally (*Eklund et al., 2017*). However, effectiveness of the lethal measures as a deterrent to reduce human-wildlife conflicts are questionable (*Peebles et al., 2013*) in addition to the negative effects of removing apex predators from an ecosystem. Government agencies have often advocated culling for certain populations of large carnivores or suggested targeted killing of problem individuals (*Inskip & Zimmermann, 2009*). Yet, non-lethal methods, have the potential to balance between the conservation of large predators and protect human property and secure livelihoods within shared landscapes (*Van Eeden et al., 2018*). Such methods are diverse and includes audio or visual deterrents, physical barriers etc. However, non-lethal methods provide the desired benefits only when local community takes ownership of the problem and participate in timely implementation of the mitigation measures (*Eklund et al., 2017*).

Human-carnivore conflicts are severe in Asia with a diversity of large carnivores i.e., tiger (*Panthera tigris*), common leopard (*Panthera pardus*), snow leopard (*Panthera uncia*), Asiatic black bear (*Ursus thibetanus*), brown bear (*Ursus arctos isabellinus*), wolf (*Canis lupus* spp), wild dog (*Cuon alpinus*) and striped hyena (*Hyaena hyaena*). Protected areas are small in this region. The region also is experiencing a rapid rise in human, livestock populations and encroachment of wildlife habitats, expansion of agricultural farms. Within such multiple-use anthropogenic landscapes large carnivores share space and resources with humans and occur in close proximity to settlements (*Naha et al., 2016*; *Naha, Sathyakumar & Rawat, 2018*). Amongst this diversity of large carnivores, human-leopard conflicts are a serious conservation problem. A major hotspot of human-leopard conflict is India. Only 5% of India's geographical area is under the protected area network and leopards occur widely throughout the country, such that leopards co-occur with humans within agro-pastoral, forested landscapes (*Karanth et al., 2009*). Such anthropogenic landscapes often lack large wild prey and leopards frequently kill livestock and domestic dogs (*Athreya et al., 2016*). Livestock depredation is a major conservation problem for the species and

attacks on humans also occur as a consequence of leopard presence near settlements or due to specific human behaviour and activity (*Jacobson et al., 2016*). A series of recent studies have also documented a rise in human-leopard conflicts in India and have examined various aspects such as nature of human-leopard relations, movement behaviour, diet, extent of self-reported livestock loss and attacks on humans (*Ghosal et al., 2013*; *Odden et al., 2014*; *Miller, Jhala & Schmitz, 2016*; *Naha, Sathyakumar & Rawat, 2018*). Some of the prominent factors influencing human-leopard conflicts are landscape features, season, time of day, availability of wild prey, livestock herd size and type of livestock (*Miller, Jhala & Schmitz, 2016*). Apart from these factors, human-carnivore conflicts are often a consequence of both human and carnivore behaviour. Animal husbandry practices, condition of livestock enclosures, location of grazing pastures close to protected areas or forested habitats and lack of animal shelters also impact the extent of predation on livestock (*Sangay & Vernes, 2008*; *Tamang & Baral, 2008*; *Khorozyan et al., 2015*; *Miller, Jhala & Schmitz, 2016*; *Broekhuis, Cushman & Elliot, 2017*). However, there are also evidence that individuals or demographic groups such as adult and older males within carnivore populations are responsible for majority of livestock depredation. Such traits could be due to the larger home ranges and ranging patterns of male carnivores, learned and risk-taking behaviour compared to females (*Odden et al., 1999*; *Farhadinia et al., 2018*).

Through this study, we evaluate the efficacy of a non-lethal visual predator deterrent (i.e., fox lights) to reduce livestock losses to leopard attacks. Pauri Garhwal district in Uttarakhand state, India, within western Himalaya has a history of human-leopard conflicts (*Goyal, Chauhan & Yumnam, 2007*) with over 160 persons injured in leopard attacks between 2006–2016. Livestock rearing is a major profession of the rural populations and losses to leopard attacks have often led to retaliatory killings. A total of 125 leopards were killed by local community members or shot dead by the district administration between 1990–2005 (*Goyal, Chauhan & Yumnam, 2007*; *Naha, Sathyakumar & Rawat, 2018*). Individual families raise cattle (*Bos taurus*), small goats (*Capra hircus*) and households often own a domestic dog. Domestic dogs are not trained guard dogs. Livestock are grazed in the forest patches; pastures during the day. These grazing lands are close to villages. Livestock are generally kept within enclosures at night. Such livestock enclosures or night shelters are made of locally available stones, mud and wood and are usually located adjacent to their houses. During the wet season, livestock are kept within enclosures and individual families provide fodder to the animals. Leopards kill livestock in grazing lands near the villages during the day and at shelters during night. Apart from making noise by beating empty canisters and some lights, villagers do not have any ways to protect their livestock from predation by leopards. Lethal control by the state government agencies is undertaken when a leopard is considered a threat to human lives and declared as a man-eater. Human-leopard conflicts are a major conservation problem in the western Himalaya (*Dar et al., 2009*; Shehzad et al., 2015; *Naha, Sathyakumar & Rawat, 2018*. Though there are reports of human-bear conflicts, they are localized within certain pockets in areas beyond 2,500-meter elevation (*Silori, 2007*). Anthropogenic mortality due to livestock depredation and attacks on humans is the primary threat to leopards in this region (*Goyal, Chauhan & Yumnam,*

*2007*). Thus we focus our study on leopard attacks taking in consideration the threats to human livelihoods and shared nature of habitats.

Depending on the size and spread of the village, fox lights were mounted at specific vantage points, at the periphery of a cluster of houses. The lights are solar-powered that flicker at random time intervals automatically during nights. These lights mimic movement or activity of local community members at the vantage points within the village. The lights are equipped with a computerised varying flash with three different colours. There are nine LED bulbs which project light at 360 degrees and can be seen over a kilometre. Fox lights have been used to deter lions from entering bomas in Kenya, elephants from crop raiding in Zambia, snow leopards from corrals in Nepal but their effectiveness are yet to be tested. Fox lights have demonstrated short-term success in reducing livestock depredation by wolves (*Canis lupus*) in US and pumas (*Puma concolor*) in Chile (*Stone et al., 2017*; *Ohrens, Bonacic & Treves, 2019*). The fox lights are equipped with 3 different coloured lights whereas lion lights have only one. However, no scientific study till date have compared effect of fox lights vs lion lights on reducing predation of livestock by large carnivores.

We hypothesize that fox lights will reduce frequency of livestock losses due to fatal leopard attacks during night. We expect that fox lights will be effective in reducing predation on livestock by leopard within open habitats as carnivores are reported to ambush prey specifically in areas with dense vegetation cover (*Ogada et al., 2003*; *Kolowski & Holekamp, 2006*; *Rostro-García et al., 2016*). We also hypothesize that improved animal husbandry practices such as condition of livestock enclosure, number of guard dogs and abundance of livestock within a site will have a significant effect on efficacy of fox lights in reducing predation  by leopards (*Ogada et al., 2003*; *Stone et al., 2017*; *Broekhuis, Cushman & Elliot, 2017*). We define a fatal attack leading to death to one or more heads of livestock (cattle, goats, sheep). Specifically, we examine (1) Effectiveness of fox lights in deterring leopard attacks on livestock (2) Identify landscape features and animal husbandry practices which increase vulnerability of livestock to leopard attacks.

## MATERIAL & METHODS

### Study area

This study was conducted with the permission and support of the Uttarakhand Forest Department. The study was conducted within the Pauri Garhwal district in Uttarakhand state, India that falls within the western Himalaya. Two protected areas, viz. Rajaji and Corbett National Parks (Tiger Reserves) fall partially within this district. This is predominantly a mountainous district with an area of 5,444 km$^2$ and is part of the lesser, middle Himalaya mountains. The elevation range lies varies between 295–3,100 m (Fig. 1). Based on the Forest Survey of India report (*Forest Survey of India, 2017*), the region has a forest cover of 64%, with the primary land cover being moderate dense forest followed by scrublands and open forests. The region is a landscape matrix of forests, scrubland, agricultural areas and human settlements. Average rainfall in the district ranges between 218–235 cm. Human population density is moderate i.e., 110 persons

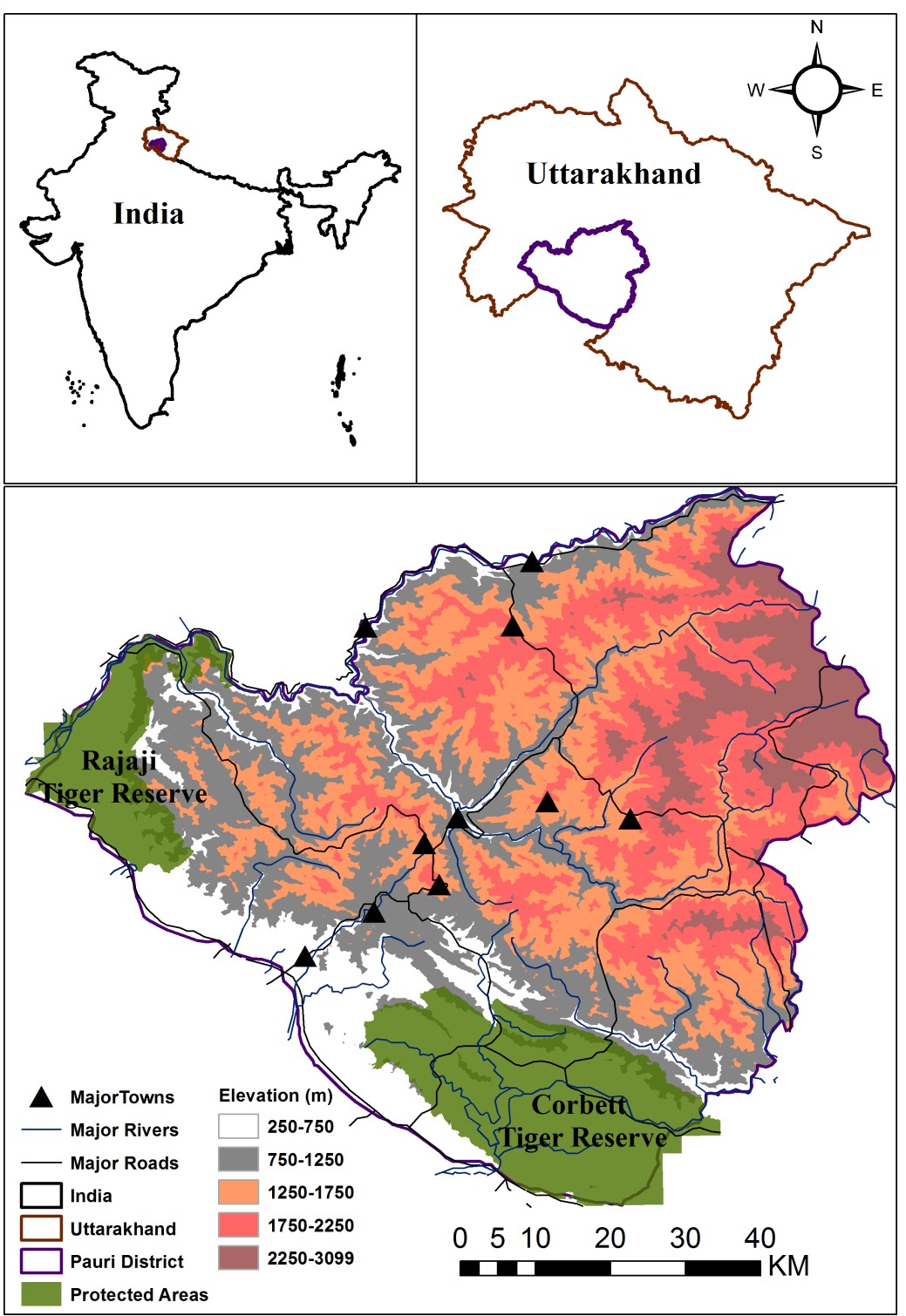

**Figure 1  Location of Pauri Garhwal District within India and Uttarakhand.** The map depicts protected areas, major roads, rivers, towns and elevation gradient within the Pauri Garhwal District.

per km$^2$ (*Census of India, 2011*). Due to outmigration, 331 villages were abandoned and the district recorded an annual growth rate of $-1.4$ percent between 2001–2011 (*Census of India, 2011*). Livelihood opportunities are limited with the major professions being livestock farming, agriculture and cottage industries. Livestock density of this region is 58 per km$^2$ (*Uttarakhand Department of Animal Husbandry, 2012*) whereas the major mammalian fauna is common leopard, Bengal tiger, Asiatic black bear, barking deer (*Muntiacus muntjak*), goral (*Naemorhedus goral*), sambar (*Rusa unicolor*), wild pig (*Sus scrofa*), rhesus macaque (*Macaca mulatta*) and common langur (*Semnopithecus entellus*) (*Goyal, Chauhan & Yumnam, 2007*).

## Data collection and experimental set up

We adopted a participatory approach to create awareness about the nature of leopard attacks, ecology, importance of large carnivores and adoption of non-lethal predator deterrents by the local community members. Participatory approaches have often been regarded as effective means to alleviate human-carnivore conflicts and implement specific interventions (*Treves, Wallace & White, 2009*). We conducted a series of conservation awareness workshops ($N = 30$) from March 2017 to March 2018 targeting local community members about the possible non-lethal interventions to reduce livestock predation by leopard, biology of leopards, role of large carnivores within ecosystems and importance of animal husbandry practices. We do not measure the efficacy of the conservation awareness programs in our current study and only focus on the performance of fox lights in reducing livestock predation by leopards. Community members ($N = 80$) who agreed to cooperate with our research team or were nominated by the village heads, were identified from this group and recognised as regional guardians. The regional guardians had some levels of formal education, intimate knowledge of the region, wildlife and experience in identifying carnivore tracks. We selected 27 villages for conducting this experiment. The regional guardians and community members were briefed about the nature, design of the experiment and use of visual predator deterrents. Selection of the experimental and control sites were done in consultation with the local forest staff, village heads and examination of compensation records regarding livestock losses to leopard attacks in the past two years. A total of ($N = 16$) locations were selected from 10 villages for setting up the predator deterrents. We selected another ($N = 17$) locations from the remaining 17 villages as control sites (Fig. 2). Three to four regional guardians were responsible for managing an experimental unit. The regional guardians were aware whether their village was part of the experiment or control site and reported any incident of malfunctioning within 4–6 h. The experiments were conducted during the period April 2018 and April 2019.

The regional guardians assisted our research team in setting up the deterrents at specific vantage points within the village such as ridgelines, rooftops, animal trails and pasture lands (Fig. 3). We installed two fox lights at two edges of an imaginary circle (50 m radius) surrounding a cluster of houses within a village. The lights were installed or mounted on iron rods high enough in order to make it visible for leopards depending on the surrounding vegetation and topography. The lights randomly emitted three different coloured flashlights and were manually activated at dusk. Lights were switched off at

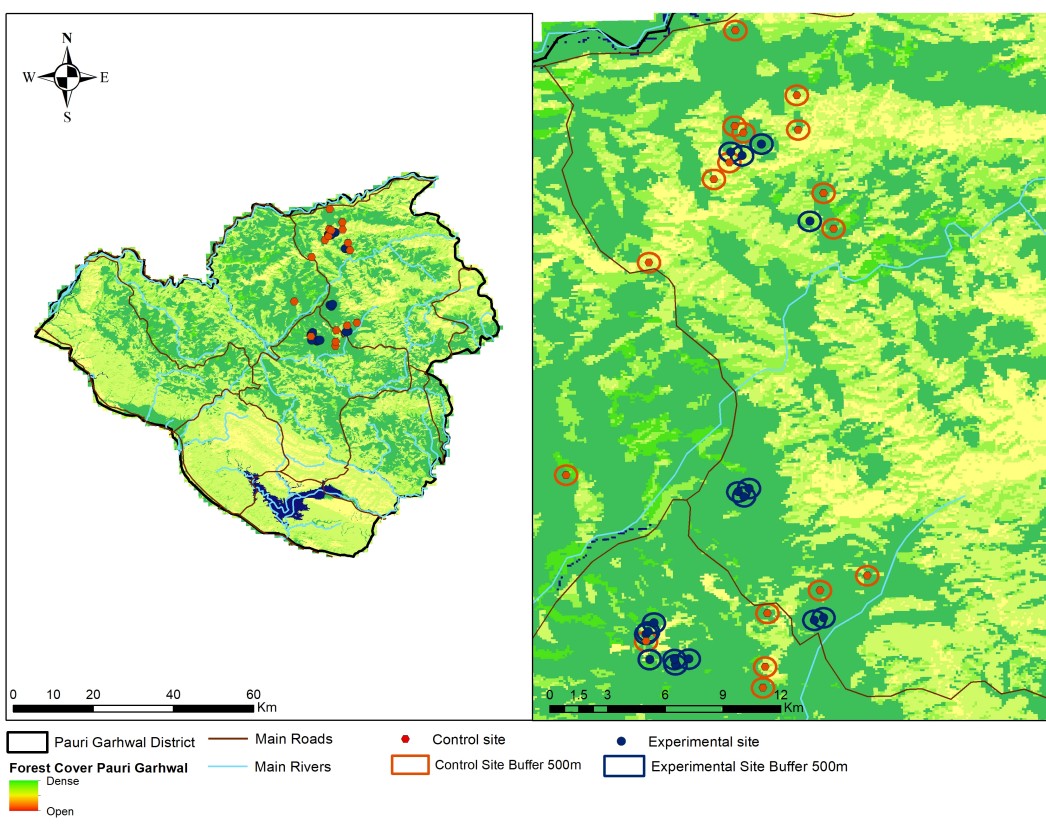

**Figure 2 Location of experimental (fox lights) and control site locations within Pauri Garhwal District.** The locations indicate experimental and control sites with buffers within Pauri Garhwal District.

dawn. To prevent habituation by leopards, all lights within the experimental sites were switched off randomly three days a week. This random pattern was decided by the regional guardians. To confirm visitation by leopards within the vicinity of the experimental and control sites, we regularly sampled trails ($N = 27$) and recorded presence of leopard pugmarks, scrape marks, scats within 50 and 500 m radius of the imaginary circle. We also consulted the regional guardians and verified presence of leopard signs and livestock predation events during the experimental period. Data on livestock depredation by leopards were collected from the experimental and control sites during the study period. Regional guardians, livestock owners and our research team members correctly identified livestock kills to leopards based on predation signs, scrapes, vocalization, throat bite and direct observations (*Karanth & Sunquist, 1995*; *Khorozyan et al., 2018*). Research team members were also trained to identify carnivore signs accurately based on the National Tiger Conservation Authority protocol (*Jhala et al., 2009*). Predation by Asiatic black bear was negligible within these villages (confirmed through wildlife compensation registers) and hence there was no ambiguity in livestock kills by leopard. We tested the efficacy of fox lights at two different spatial scales and collected data on livestock depredation by common leopard from experimental sites ($n = 16$) and control sites ($n = 17$) for a period of one year.

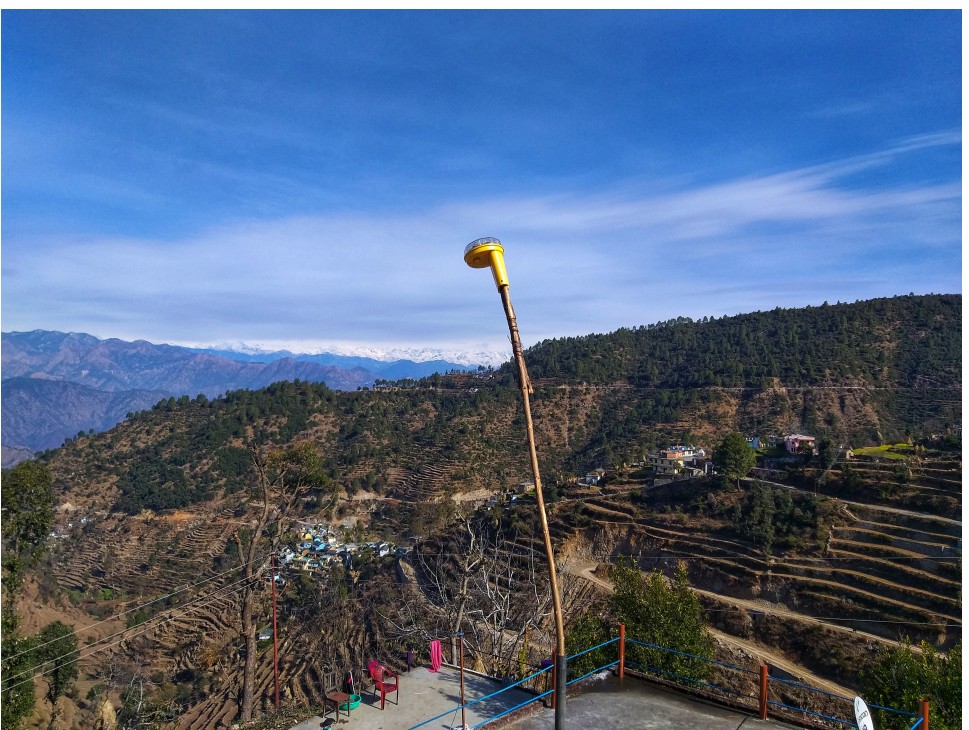

**Figure 3** Image of a fox light deployed by regional guardians and researchers at the periphery of human settlements within a village in the Himalaya.

## Analyses

We ran 3 analyses at 2 spatial scales to examine the impact of sociological variables, landscape features, fox lights and animal husbandry practices on livestock depredation by leopards. Spatial scale of predator and prey decision making changes throughout the hunting process affecting the probability of predation (*Hilborn et al., 2012*). Certain landscape features and sociological variables influence the outcome of such processes. Considering that decision making for livestock kills by large carnivores occurs at both finer and coarser spatial scales, we considered 2 different scales (50 and 500 m) for assessing vulnerability of predation (*Miller, Jhala & Jena, 2015*; *Amirkhiz et al., 2018*). We recorded data for seven socioecological variables within a 50-m circle of the experimental and control sites. The socio-ecological variables include: number of households, total number of people, condition of livestock enclosure, number of livestock, total number of guard/domestic dogs, vegetation cover (percentage of herb, shrub, tree and barren land) and altitude. Altitude was measured using a GPS whereas vegetation cover was estimated visually using a 50 m radius circular plot while the other details were recorded through a questionnaire survey (Appendix S1).

To explore the effect of ecological predictors, we generated individual buffer of 500 m radii around control and treatment sites using Arc GIS 10.3.3. For each of these circles, we generated information for six important landscape variables based on their ecological importance such as landscape features (area of non-forest, open forest, moderate dense

forest, dense forest), topographic features (altitude) and intensity of nightlight. Vegetation and human presence were regarded as major predictors of large carnivore predation on livestock (*Ugarte, Darío & Javier, 2019*). In our previous study on leopard fatal attacks on humans in the Pauri Garhwal region (*Naha, Sathyakumar & Rawat, 2018*), presence of river/water bodies was not identified as a significant predictor and hence we discarded distance to rivers/presence of rivers as a variable for our analyses. Since our study area was confined to a small region within the district, We did not consider topographic complexity such as slope, aspect and instead included altitude as an influential variable. Altitude was identified as a major predictor of leopard attacks on humans in this region (*Naha, Sathyakumar & Rawat, 2018*). We extracted the mean altitude value for each site (control and treatment) based on digital elevation maps with 90-m spatial resolution. We were also interested in examining broader seasonal patterns of depredation (dry and wet) and not just for individual months, hence the experimental period was divided into 2 primary seasons (Dry –April–June, November–March, Wet –July–October).

Landscape features- Probability of livestock depredation by large carnivores were linked to several features such as type of vegetation, human infrastructure/presence and altitude (*Miller, Jhala & Jena, 2015*).

1.  We hypothesized that predation risk by leopard would be higher in sites with moderate to dense forests/vegetation cover (*Miller, Jhala & Jena, 2015*; *Rostro-García et al., 2016*). We calculated landscape variables for each site, i.e., area under different land-use types from forest type map of India (*Forest Survey of India, 2017*).

2.  Human presence- We hypothesized that leopards would avoid killing livestock in areas with increased human presence (*Rostro-García et al., 2016*). We extracted night light values using the 1,000-m spatial resolution night-time visible light data of India.

3.  Altitude- Considering that livestock killing by carnivores have a positive relationship with altitude (*Kissling, Fernandez & Paruelo, 2009*; *Zarco-Gonzalez, Monroy-Vilchis & Alanız, 2013*; *Miller, 2015*; *Rostro-García et al., 2016*), we hypothesized that predation risk by leopards would be higher in elevated regions.

## Data preparation and analysis

Once data were compiled, we prepared master tables for the 2 spatial scales (50 and 500-m radius circles) (Tables 1 and 2). We did Pearson correlation and omitted all correlated variables $\geq 0.70$ (*Dormann et al., 2007*) using R version 3.4.0 (R Development Core Team 2017). We prepared count statistic data for the number of livestock predation events recorded within a site. We assigned 0 to sites that had no attacks. We used generalized linear mixed models (GLMMs) with Poisson structure and village name as random factors nested within sites with and without deterrents/fox lights to quantify effect of predictor variables. We considered (habitat type, human presence, altitude) for 500 m radius circles, vegetation cover (altitude and proportion of shrub, herb, tree and barren land) for 50 m radius circles, fox lights and modelled probability of livestock predation by leopard. For the Poisson structure, our response variable was the number of livestock killed by leopards at night within each individual cluster during the experimental period. Livestock predation by leopard could potentially be different within villages/localities and also within sites

Naha et al. (2020), *PeerJ*, DOI 10.7717/peerj.9544

**Table 1  Major predictor variables considered for regression analysis within a fine scale of 50- 2 m radii of experimental and control sites in Pauri Garhwal.**

| Type of variable | Predictor variable | Unit | Resolution | Source |
|---|---|---|---|---|
| Habitat (Landscape variables) | Proportion of herb cover | Percentage | 50-m radii | Recorded during field survey |
| | Proportion of shrub cover | Percentage | 50-m radii | Recorded during field survey |
| | Proportion of barren land cover | Percentage | 50-m radii | Recorded during field survey |
| | Proportion of tree cover | Percentage | 50-m radii | Recorded during field survey |
| Altitude | DEM | M | 50-m radii | Recorded during field survey |
| Livestock husbandry practices | Number of household | Numeric | 50-m radii | Recorded during field survey |
| | Number of people | Numeric | 50-m radii | Recorded during field survey |
| | Number of livestock | Numeric | 50-m radii | Recorded during field survey |
| | Enclosure type | Categorical | 50-m radii | Recorded during field survey |
| | Number of domestic guard dogs | Numeric | 50-m radii | Recorded during field survey |
| Livestock lost to leopard attacks | Number of livestock killed in forest patch | Numeric | Vicinity of village (500-m radii) | Recorded during field survey |
| | Number of livestock killed within enclosure | Numeric | 50-m radii | Recorded during field surveys |
| Deterrent | Presence of fox light | Binary/Factor | 50-m radii | Recorded during field surveys |

**Table 2  Major predictor variables considered for regression analysis within a broader scale of 500-m radii of experimental and control sites in Pauri Garhwal.**

| Type of variable | Predictor variable | Unit | Resolution | Source |
|---|---|---|---|---|
| Habitat (Landscape variables) | Area of non-forests | $m^2$ | 30 m | FSI, 2017 |
| | Area of scrubland | $m^2$ | 30 m | FSI, 2017 |
| | Area of moderate dense forests | $m^2$ | 30 m | FSI, 2017 |
| | Area of very dense forests | $m^2$ | 30 m | FSI, 2017 |
| | Area of open forest | $m^2$ | 30 m | FSI, 2017 |
| Human presence and infrastructure | Night light | Radiance | 500-m radii | Census India, 2011 |
| Altitude | DEM | M | 90 m | DEM |
| Deterrent | Presence of fox light | Binary/Factor | 500-m radii | From field survey |

with or without deterrents. Variation in predation due to different localities/villages were considered as a random error in the model. We used location/village name (1–27) and presence of deterrent/fox light (presence of fox light: 1, absence of foxlight: 2) within a site as categorical factors in the analysis. The analysis was done in R using the function glmer in the package lme4 (*Bates et al., 2012*). The proportion of barren land cover was negatively correlated ($-0.75$) with proportion of shrub cover, hence we removed barren land cover from the analysis.

## Livestock husbandry

To model livestock losses as a function of animal husbandry practices and presence/absence of fox lights we used the same response variable used for identifying landscape predictors of predation risk within a fine scale of 50-m circle. We used generalized linear mixed models (GLMMs) with a Poisson structure and considered sociological variables (household size, number of houses), animal husbandry practices (condition of livestock enclosure, number of livestock, number of guard dogs), location (village name) and presence of fox lights. We used location/village name as random factors nested with sites with and without deterrents. Village name (1–27) and presence of fox lights (presence of fox light: 1, absence of foxlight: 2) were considered as categorical factors in the analysis. To determine the condition of livestock enclosure we considered strength of the construction materials in the following order (categorical: branches-1, wooden poles-2, stone walled-3, cemented-4). The analysis was done in R using the function 'glmer' within the package 'lme4' (*Bates et al., 2012*).

We used a priori candidate models and ranked them based on AIC values. Models with the lowest AIC values for all 3 analyses were considered the best or dominant model (*Burnham & Anderson, 2002*) and the output (coefficients and estimates) explained the probability of livestock predation by leopards within IHR. We averaged parameter estimates across models with AIC differences ($\Delta$ AIC $< 2$) (*Burnham & Anderson, 2002*).

We checked for diurnal livestock attacks after installation of the lights between experimental and control sites using chi-square test in R. We used Wilcoxon-Signed-Rank Test and chi-square test to check for presence of leopard signs, effectiveness of fox lights in deterring attacks, difference in temporal, seasonal patterns and type of livestock killed between experimental and control sites. Since data was not normally distributed, we also

compared predictor variables between the experimental and control sites using Wilcoxon Signed-Rank Test in R. Statistical significance was $P \leq 0.01$ for all analyses. All spatial analyses were performed with Arc GIS 10.3.3 and R.

## RESULTS

### Livestock depredation within control and experimental sites

We confirmed presence of leopards within the vicinity of the experimental and control sites through trail walks (43 leopard signs i.e., pugmarks) and secondary information (4 sightings and 19 signs i.e., pugmarks) during the study period. However, there was no significant difference in leopard signs, secondary information between experimental and control sites ($W = 168.5, p = 0.237$). A total of 105 livestock were killed by leopards within 10 (out of 27 sites) villages of the Pauri Garhwal district during the study period. A total of 47% of the livestock killed within experimental and control sites were goats, 37% were cows and the rest were calves ($\chi^2 = 16.24, df = 1, p < 0.01$). Livestock predation was higher (56%) during the dry season when compared to the wet season ($\chi^2 = 1.44, df = 1, p > 0.01$).

We recorded 36 (34%) and 69 (66%) livestock kills within experimental and control sites respectively ($\chi^2 = 10.24, df = 1, p < 0.05$). A total of 33 cases (92%) of the total livestock kills within experimental sites and 64 cases (93%) of the total livestock kills within control sites occurred outside livestock enclosures. Out of the total 105 livestock kills, 63 (60%) occurred during daylight and the remaining occurred during night ($\chi^2 = 4, df = 1, p < 0.01$). There was significant difference in temporal pattern of livestock depredation between experimental and control sites ($\chi^2 = 17, df = 3, p < 0.01$). Within experimental sites, 25 (70%) of the predation events occurred during day and the remaining occurred during night ($\chi^2 = 16, df = 1, p < 0.01$). There was no evidence for any temporal pattern of leopard depredation within control sites ($\chi^2 = 1, df = 1, p > 0.01$). The average proportion of vegetation cover within experimental sites (50-m) was estimated to be herb (14.68%), shrub (50.31%), tree (17.81%) and barren land cover (17.18%) whereas for control sites it was herb (16.17%), shrub (42.35%), tree (23.53%) and barren land cover (17.35%) respectively. There was no significant variation in vegetation cover between experiment and control sites ($\chi^2 = 1.5, df = 3, p\ 0.672$).

### Characteristics of control and experimental sites

An average of 26 livestock (SE 21), range (3–120) were present within a cluster of 50-m circle. The average elevation of experimental and control sites was 1,533 m (SE 148), range (1,086–1,823). The average number of people staying within a cluster was estimated to be 17 members (SE 4), (range 5–30) whereas the average number of houses was 7 (SE 2), (range 1–18). Households possessed an average of 1 guard dog (SE 1), (range 0–4). About 42% of the livestock enclosures were made of wooden poles, 36% branches, 12% stones and 10% were cemented. Wilcoxon signed rank sum test results indicate that none of the predictor variables (at 50 or 500 m radii) differed significantly between experimental and control sites.

**Table 3** Second-order Akaike Information criterion scores (AIC), ΔAIC of generalized linear mixed models with Poisson structure predicting livestock depredation by common leopards in Pauri Garhwal within a fine scale of 50 m radius around human settlements.

| Model Number | Model | AIC | ΔAIC |
|---|---|---|---|
| 1. | Presence of fox light + Proportion of scrub cover | 95.27 | 0 |
| 2. | Presence of fox light + Proportion of herb cover + Proportion of scrub cover + Proportion of tree cover | 95.56 | 0.29 |
| 3. | Presence of fox light + Proportion of herb cover + Proportion of tree cover | 96.14 | 0.87 |
| 4. | Presence of fox light + Proportion of herb cover | 96.74 | 1.47 |
| 5. | Presence of fox light + Proportion of herb cover + Proportion of scrub cover + Proportion of tree cover + Altitude | 97.53 | 2.26 |

## Influence of landscape predictors on livestock depredation by leopards

On a fine scale (50m radii), presence/absence of fox light was the best predictor of livestock depredation by leopard (Table 3, Table S1). Leopards were most likely to kill livestock in areas with no fox light (estimate −1.067, CI [−0.34019−1.79492]). After accounting for the effect of village name/localities nested within sites with and without fox lights as a random error, we found no significant effect of herb, shrub, tree and altitude on livestock predation by leopard (Table S1).

On a coarser scale of 500-m radius, there were no significant landscape predictors of leopard attacks on livestock (Table 4). The effect of altitude, night light, non-forest, scrubland, open forest, moderate dense forest and very dense forest displayed a weak positive relationship with probability of livestock depredation but these were not statistically significant (Table S2). There was no significant effect of fox light on livestock predation by leopard (Table S2).

## Livestock husbandry

The model averaged coefficients indicates that nocturnal livestock depredation events had a positive relationship with the number of household, number of guard dogs and enclosure type whereas it displayed a negative relationship with number of people and livestock present within a 50-m circle of human settlements (Table 5, Table S3). Likelihood of a depredation event within a 50-m cluster was higher in sites with houses and domestic guard dogs. After accounting for the effect of village name/localities nested within sites with and without fox lights as a random error, we found significant effect of fox lights on livestock predation by leopard (−0.96264 CI [−0.14991−1.77537]). The likelihood of livestock depredation was lower within a site with the presence of fox lights.
**Table 4  Second-order Akaike Information criterion scores (AIC), ΔAIC of generalized linear mixed models with Poisson structure predicting livestock depredation by common leopards in Pauri Garhwal within a coarser scale of 500 m radius around human settlements.**

| Model Number | Model | AIC | ΔAIC |
|---|---|---|---|
| 1. | Presence of fox light + Area of scrub + Area of very dense forest | 103.8 | 0 |
| 2. | Presence of fox light + Nightlight + Area of scrub + Area of open forest + Area of moderate dense forest + Area of very dense forest | 106.7 | 2.9 |
| 3. | Presence of fox light + Nightlight + Area of scrub + Area of open forest + Area of very dense forest | 107.4 | 3.6 |
| 4. | Presence of fox light + Nightlight + Area of non-forest + Area of scrub + Area of open forest + Area of moderate dense forest + Area of very dense forest | 108.1 | 4.3 |
| 5. | Presence of fox light + Altitude+ Nightlight + Area of non-forest + Area of scrub + Area of open forest + Area of moderate dense forest + Area of very dense forest | 110 | 6.2 |

**Table 5  Second-order Akaike Information criterion scores (AIC), ΔAIC of generalized linear mixed models with Poisson structure for influence of livestock husbandry on probability of livestock depredation by common leopards within a fine scale of 50 m radius around human settlements.**

| Model Number | Model | AIC | ΔAIC |
|---|---|---|---|
| 1. | Presence of fox light + Enclosure type Deterrent | 100.8 | 0 |
| 2. | Presence of fox light + Enclosure type + Number of domestic guard dog | 101.2 | 0.4 |
| 3. | Presence of fox light + Enclosure type + Number of livestock + Number of domestic guard dog | 103.2 | 2.4 |
| 4. | Presence of fox light + Number of household + Enclosure type + Number of livestock + Number of domestic guard dog | 104.7 | 3.9 |
| 5. | Presence of fox light + Number of household + Enclosure type + Number of livestock + Number of domestic guard dog | 106.6 | 5.8 |

# DISCUSSION

Our study provides evidence-based results to manage large carnivores within human-dominated landscapes and highlights effectiveness of non-lethal visual deterrents to reduce livestock depredation. This study is the first known experiment testing the effectiveness of non-lethal visual deterrents in reducing livestock losses to common leopards in South Asia. We found that flashlight devices deterred predation by leopards on livestock. There was significant decline in livestock predation by leopard but no difference in leopard visitation or presence between experimental and control sites. Significant decline in livestock depredation by leopard in sites with predator deterrents support the hypothesis that fox lights reduced the number of livestock losses to nocturnal leopard attacks within villages in the western Himalaya. However, we found no support for the hypotheses

regarding the influence of vegetation cover, open habitats, animal husbandry practices, moderate/dense forests, human presence and altitude on the probability of livestock depredation by leopard. Hence we assume that livestock depredation by leopards were random in nature with respect to the variables measured. Predation on livestock is the stimuli for human-carnivore conflicts globally and such events have to be addressed effectively to ensure survival of large carnivores within human-dominated landscapes. Though we did not measure the effectiveness of our community based conservation programs, our results suggest the potential for adopting non-lethal visual deterrents through involvement of the local community members in reducing livestock losses to large predators across heterogeneous landscapes of South Asia.

Our results suggest that there was no significant influence of environmental variables on livestock predation by leopards. Previous studies have documented that large carnivores such as stalking hunters i.e., tigers, jaguar (*Panthera onca*), pumas, lions and leopards use dense vegetation cover and forested habitats to hunt prey (*Inskip & Zimmermann, 2009*; *Kissling, Fernandez & Paruelo, 2009*; *Zarco-Gonzalez, Monroy-Vilchis & Alanız, 2013*; *Miller, Jhala & Jena, 2015*; *Broekhuis, Cushman & Elliot, 2017*; *Khorozyan et al., 2017*). A study conducted in eastern Himalaya documented that risk of leopard killing livestock increased with forest cover (*Rostro-García et al., 2016*). Altitude has been reported to be a significant predictor of livestock predation for jaguars, pumas and leopards especially in high elevated areas of Mexico, Argentina and Bhutan (*Kissling, Fernandez & Paruelo, 2009*; *Zarco-Gonzalez, Monroy-Vilchis & Alanız, 2013*; *Rostro-García et al., 2016*). On the contrary, we did not document any significant relationship of predation risk and altitude within our study sites. Human presence and infrastructure have been reported to have a negative relationship with predation risk by large carnivores (*Miller, 2015*). We did not document any significant effect of human presence on predation risk by leopards. Such results could be due to the relatively low sample size of only 27 villages with minimal variation in topography, land use patterns and human presence.

We also found that sixty-percent of the livestock killings were diurnal in nature which is contrary to previous findings from western and eastern Himalaya i.e., Pakistan and Bhutan where they were nocturnal (*Sangay & Vernes, 2008*; *Dar et al., 2009*). Radio-telemetry studies in Nepal and India have documented leopards to be nocturnal (*Odden & Wegge, 2005*; *Odden et al., 2014*) but our results suggest diurnal activity peaks within human dominated mountainous landscapes. Cheetahs and lions in eastern Africa (*Broekhuis et al., 2014*; *Lesilau et al., 2018*) and tigers in Sundarban delta (*Naha et al., 2016*) have also been reported to exhibit diurnal activity peaks and are believed to be the major driver of human-carnivore conflicts. Leopards probably prefer to kill wild prey at night whereas livestock killing is diurnal due to the availability, poor or unsupervised grazing practices, and ease of catching domestic prey. We also did not document any significant seasonal variation in leopard attacks on livestock which was similar to studies conducted in Iran (*Khorozyan et al., 2017*).

Livestock husbandry practices have been reported to affect the likelihood of leopard predation on livestock in mountainous regions of South Asia (*Dar et al., 2009*; *Kabir et al., 2014*; Shehzad et al., 2015; *Khorozyan et al., 2017*). We found no evidence of animal practices

impacting predation events by leopards in Pauri Garhwal. Improving condition of animal enclosures, use of livestock guardians (herders and trained dogs), visual, auditory deterrents and lethal control of predators have been identified as the major interventions which have effectively reduced livestock losses (*Van Eeden et al., 2018*; *Miller, Jhala & Schmitz, 2016*; *Eklund et al., 2017*). Light based deterrents have been documented to effectively protect livestock against lions (*Panthera leo*) (*Lesilau et al., 2018*) and pumas (*Ohrens, Bonacic & Treves, 2019*) and our results also support such findings. However, not all visual deterrents are effective, e.g., scarecrows and lion lights have failed to prevent livestock losses to leopard attacks in east Africa (*Broekhuis, Cushman & Elliot, 2017*).

Fortified and improved enclosures have been largely documented to be effective in reducing livestock losses to multiple predators such as wolves, pumas, spotted hyenas and lions in Europe, South America and Africa (*Lichtenfeld, Trout & Kisimir, 2015*; *Eklund et al., 2017*; *Van Eeden et al., 2018*). Yet such measures have not provided success in deterring leopard attacks on livestock in Africa (*Eklund et al., 2017*). Several studies have documented that herd size in a village is directly proportional to the number of predator attacks (*Van Bommel et al., 2007*; *Woodroffe et al., 2007*). However, we did not find any evidence of a significant relationship between the number of livestock present within a cluster of settlements and probability of livestock depredation by leopards.

It is important to reduce livestock losses but perceived risk towards large predators are also influenced by a combination of several social, cultural variables (*Dickman, 2010*). Community based-conservation programs are successful when local members are directly involved and take ownership of the project. We demonstrate that it is possible to overcome challenges within a semi-natural ecosystem such as a village society by having moderate control over recruitment of participants and recognizing community leaders. By adopting a community-based conflict mitigation approach we have been successful in reducing human-leopard conflicts within a multi-use landscape of the Himalayan region. Similar success stories such as the ''Lion Guardians'' project in east Africa (*Hazzah et al., 2014*), snow leopard community based conservation programs in India (*Vannelli et al., 2019*), Tiger Team initiative in Bangladesh Sundarbans (*Inskip et al., 2016*) and Persian leopard conservation project in Iran (*Khorozyan et al., 2017*) have demonstrated considerable success in improving human-predator relations and created pathways of coexistence within developing regions of the world.

We acknowledge some limitations of our study. The first is regarding the small sample size of villages, localized nature of the study and random operation of the fox lights adopted by the local community members. Second, we did not have data on leopard density or occupancy for the region nor did we have information on abundance of wild prey. Third, we could not measure the effect of habituation and behavioural response of leopards towards fox lights. In spite of these limitations our study emphasizes the effectiveness of visual predator deterrents in mitigating human-leopard conflicts in South Asia.

Human-leopard conflicts are a major threat to survival of leopards outside protected areas in Asia and Africa (*Jacobson et al., 2016*). Successful implementation of conservation programs will need a coordinated effort from all multiple agencies, which includes (local communities, wildlife staff, police, civil administration, animal husbandry, agriculturists,

veterinarians, conservationists etc.). Future studies should be taken up to understand the behavioural response and habituation of fox lights and other visual deterrents to leopards in reducing attacks on livestock within multiple-use landscapes. Studies should be conducted to evaluate effectiveness of non-lethal deterrents under varying conditions within multi-predator communities globally.

Rising anthropogenic impacts affect survival of large carnivores globally and hence they are forced to occupy heterogeneous shared landscapes where persecution due to real or perceived threats to human interests or livelihoods are high (*Carter & Linnell, 2016*). To maintain coexistence within such shared landscapes, it is essential to develop conservation models which can balance human livelihoods, incorporate traditional knowledge, reduce financial losses to predators as well preserve biodiversity (*Carter & Linnell, 2016*). We provide rigorous scientific evidence that non-lethal interventions are effective in reducing predation on livestock within multiple-use landscapes of South Asia. Although, there might be differences within natural and social systems our community based approach has the potential to reduce livestock losses to similar large bodied carnivores such as jaguars, hyenas, cheetahs, tigers, snow leopards, lynx, wild dogs, wolves and bears. By reducing financial loss, we hope to ensure survival of large carnivores and thereby preserve functionality of natural ecosystems. Such measures will have cascading effects on the larger human society through flow of ecosystem services, increased wildlife tourism based livelihoods and improved human-wellbeing, safety (*Ripple et al., 2014*).

## CONCLUSIONS

We provide evidence for the effectiveness of fox lights in deterring leopard attacks on livestock in western Himalaya., Our work can be successfully replicated to reduce human-carnivore conflicts within other mountainous regions of Asia. However, conflict mitigation measures which might work at a particular place might not be successful elsewhere due to variation in animal behaviour and environmental or social factors. The majority of predator deterrent experiments are usually not successful as long-term solutions to reduce livestock depredation by large carnivores. Hence, they should be integrated with efficient animal husbandry practices, multi-interest group collaborations and community based conservation programs. Given the positive effect of these flash lights to reduce livestock depredation at night, we recommend avoidance of areas close to forest patches for grazing and supervision by an experienced herder to reduce economic losses to leopard attacks during the day.

## ACKNOWLEDGEMENTS

We thank the Divisional Forest Officer, Garhwal Forest Division, forestry staff, village community heads and non-governmental organization members for their support during fieldwork.

### Funding
This work was funded by the NMHS Large Grant program (Moef&CC) (NMHS/LG-2016/009), Govt. of India. The funders had no role in study design, data collection and analysis, decision to publish, or preparation of the manuscript.

### Grant Disclosures
The following grant information was disclosed by the authors:
NMHS Large Grant program (Moef&CC): NMHS/LG-2016/009.

### Competing Interests
The authors declare there are no competing interests.

### Author Contributions
- Dipanjan Naha conceived and designed the experiments, performed the experiments, analyzed the data, prepared figures and/or tables, authored or reviewed drafts of the paper, and approved the final draft.
- Pooja Chaudhary performed the experiments, analyzed the data, authored or reviewed drafts of the paper, and approved the final draft.
- Gaurav Sonker performed the experiments, analyzed the data, prepared figures and/or tables, and approved the final draft.
- Sambandam Sathyakumar conceived the experiment, authored or reviewed drafts of the paper, supervision of the project, and approved the final draft.

### Field Study Permissions
The following information was supplied relating to field study approvals (i.e., approving body and any reference numbers):

The Chief Wildlife Warden of Uttarakhand granted permission for research in Pauri Garhwal.

### Data Availability
Raw data is available in the Supplementary Files.

### Supplemental Information
Supplemental information for this article can be found online at http://dx.doi.org/10.7717/peerj.9544#supplemental-information.

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
