# Peer review of "Effectiveness of non-lethal predator deterrents to reduce livestock losses to leopard attacks within a multiple-use landscape of the Himalayan region"

_PeerJ, doi:10.7717/peerj.9544_

## Round 0.1 · original submission · Major Revisions

My two reviewers have provided very detailed comments. I know as an author how helpful this is when revising a manuscript. In addition, Andrew Jacobson has identified himself and told me in a personal note that he would be happy to send you a Word file with his detailed comments on it. I urge you to contact him

Reviewer 1 ·

Basic reporting

Effectiveness of non-lethal predator deterrents to reduce livestock losses to leopard attacks within a multiple-use landscape of Himalaya
This is a nice and important piece of research done on leopards, using fox lights in India. The authors nicely compare the fox light efficiency in northern India and provide some amazing stats on their performances. I think the paper has the potential for using in conservation biology.
However, I have a few major concerns which I recommend to be considered in order to enhance the paper clarity and readership:
1. The paper is exhaustively long, I had to finish the revision in a few sessions as I was not able to finish once. That will cause fewer readerships. There are redundant parts particularly in Discussion, which could be merged or removed. The discussion is already 4 pages with single line, and towards the end, it is becoming of more general recommendation to reduce conflict. In Results also too many numbers have been provided, while some should be summarized in tables, some do not make any difference if removed. So, my first advice is make your paper shorter and crispy
2. My second concerns is about stats, I was wondering why the authors did not use GLMM? Based on the maps they provided, it seems that some villages are next to each other, so the paper can enjoy a GLMM analysis with nested structure of random effect, as auto correlation is likely in these data. Also, many Wilcoxon tests to compare means between control and experiment could be replaced with a single logistic regression with binary response variable. Also, it was not clear if model averaging between top scoring models was done.
3. The paper has nicely highlighted its applicability to south Asian mountainous contexts, but is lacks proper connection with previous papers published on human-leopard conflict in south Asian mountains as well as India. I have highlighted a few in my suggestions. In current form, the paper has tried to compare its findings in the context of African carnivores or other large carnivores like puma or tigers. Instead it is desirable to first compare the results with other Asian leopard contexts, as they are expected to be the most relevant ones, particularly from mountainous regions.
Minor comments
L70 Latin names are needed when first appear in the ms
L87-92: All causes of conflict mentioned here are related to the environment and people. However, there are strong evidences showing that some features of leopard individuals are also responsible for conflict. I suggest the authors to enrich their arguments using these two papers. The first paper highlights the notion of “problem animals”, showing individuals disproportionally killing livestock. The second paper applied this notion to leopards using satellite telemetry, showing old age can be a motif for livestock depredation
Odden, J., Smith, M.E., Aanes, R. and Swenson, J.E., 1999. Large carnivores that kill livestock: do" problem individuals" really exist?. Wildlife Society Bulletin (1973-2006), 27(3), pp.698-705.
Farhadinia, M.S., Johnson, P.J., Hunter, L.T. and Macdonald, D.W., 2018. Persian leopard predation patterns and kill rates in the Iran–Turkmenistan borderland. Journal of Mammalogy, 99(3), pp.713-723.
L114: It is more related to study design, not here
L115-120: It would be more desirable to have some a priori predictions on the effect of husbandry practices and landscape features on the performance of fox lights.
L134-138 These latin names must follow the species name when first appear
L190-192 Shouldn’t this be the number 1? To be followed by two more hypotheses?
L196 The role of altitude is not clear as why higher areas are more prone to livestock depredation? There is another paper showing that leopard kill livestock in lower elevations:
"Vertical relief facilitates spatial segregation of a high density large carnivore population." Oikos (2019).
L202 R should be cited properly (R Core Development …).
L202-204 Not clear what is count and what is binary? Kills are binary and number of livestock are count? Clarify please. Towards the end of this paragraph, I did not understand what the binary data was used for, as you used Poisson GLMM for count data.
L234 R version mentioned three times, just once with proper citation is enough
L244 Why “about”?
L248 p value better to be as p>0.001
L237-250 All cases are described in a way that they all seem statistically significant, while the text should reveal which one has enough evidence and which one does not have. For example, the last sentence of the paragraph is better to be modified that there was no evidence for any temporal pattern of leopard depredation in control sites (X2,…). Please seek advice of similar papers to see how these results should appear
L247-250 Your research question here is Do day/night time depredation differ between control and experiment sites? If yes, you need to run one X2 test for both, not two independently as it related to a different research question. Please take this advice for other X2 tests between control and experiment sites
L251 Any dog killed by leopards?
L251-252 Are these number in related to the livestock abundance in those villages, or you have expected equal portions for each type of livestock? If the latter, that is wrong
L251-260 Please consider providing SD or SE for all means.
L258 Is this data needed? If not, please remove redundant data as your paper is already full of data.
L285 what is range after N here?
L285-286 So this is the key message of this paper, which shows no significant difference occur between control and experiments. So, your narration implies that there is a significant difference, while your stats do not support it. Please consider rewording and make sure that the reader gets that no evidence is available for different patterns between control and experiment.
L331 Latin name if this is their first appearance
L340 In your results, you did not present any data on dog depredation, therefore all these discussion about dogs as an attractant for leopards can be speculative. Please clarify if no dog was killed, and if that is true, please be careful with these speculations.
L344 Jacobson’s paper is on the global status of leopards, they have not collected any field data on leopard feeding ecology. Instead these papers can be used as they have obtained first hand data on dog predation by leopards in south Asia
Athreya, V., Odden, M., Linnell, J.D., Krishnaswamy, J. and Karanth, K.U., 2016. A cat among the dogs: leopard Panthera pardus diet in a human-dominated landscape in western Maharashtra, India. Oryx, 50(1), pp.156-162.
Shehzad, W., Nawaz, M.A., Pompanon, F., Coissac, E., Riaz, T., Shah, S.A. and Taberlet, P., 2015. Forest without prey: livestock sustain a leopard Panthera pardus population in Pakistan. Oryx, 49(2), pp.248-253.
L360 Talking about attractant, you mention that livestock loss is in proportion to herd size, therefore the larger herds lose more animals to leopards. I presume we cannot consider this as attractant because when an animal is killed higher than its proportion is when we can call it attractant. Simply leopards have killed more in correlation with the sheep abundance.
L415-416 Why do not you guarantee? Clarify please
L422 what do you mean by “better”?
Table 3: Why both AIC and AICc? One should be enough
Table 3 and 4, in Results, model numbers have been mentioned but here there is no model number.

Experimental design

My second concerns is about stats, I was wondering why the authors did not use GLMM? Based on the maps they provided, it seems that some villages are next to each other, so the paper can enjoy a GLMM analysis with nested structure of random effect, as auto correlation is likely in these data. Also, many Wilcoxon tests to compare means between control and experiment could be replaced with a single logistic regression with binary response variable. Also, it was not clear if model averaging between top scoring models was done.

Validity of the findings

Please see above

Additional comments

Please see the above

·

Basic reporting

There is a lot of good science and writing in the MS. The work is commendable. The writing is professional and has excellent English.
Writing: There are issues with organization and clarity, and I have attempted to suggest solutions in the edited document.
Context/bibliography: References are generally good with only minor corrections suggested. Context is pretty good too, although more needs to be done in discussing 1) visual deterrents, and 2) herding/livestock keeping practices in the region.
Figures & tables; Raw data is shared and understandable. Tables are generally good - small suggestions noted. Figure 2 needs substantial work however - poor choice of colors (blue should be river for instance and too many similar colors), and there are two maps but only one scale.
Self-contained: Needs work. The results are not linked back to hypotheses at all. No mention is made of hypotheses after Methods section.

Experimental design

This is original research with aims and scope that are appropriate for PeerJ.

The research question is well defined, relevant and meaningful, and would fill a knowledge gap. However, there are multiple analyses within the MS and these need to be more carefully explained.

Investigation appears rigorous and performed ethically.

Methods are described well. But more information is needed in a few places (see edited word documents).

Validity of the findings

Validity of findings is where the gravest issues with the MS are. The underlying data is available.

However, the authors greatly overstate their findings. As far as I could understand, only one analyses actually tested the impact of hte deterrent in reducing livestock depredation, and it found the deterrent was not statistically significant. This was hidden in last paragraph of Results. Yet the Discussion and Conclusion are written as if the effectiveness of fox lights in this manner was unequivocally proven; they were not. The paper needs substantial re-working to provide better clarity about what their results state, and to not give conclusions that are not supported by their evidence. So the abstract, Discussion and conclusion need re-writing beyond what I've commented on in the edited document itself.

Additional comments

This is commendable science, important conservation work in a difficult poorly-studied region, and quite well-written. However, the major caveat is that the primary conclusion on the efficacy of fox-lights is not supported by the data or analysis I saw. The authors need to re-write the MS to ensure they are not overstating the importance of their work, or pushing an agenda rather than sticking to the results they obtained. That is the primary issue with this MS.

I have provided a lot of comments and suggested edits in the document, to assist with grammar and clarity. I hope these are helpful.

A few additional comments that need clarity and may not have been mentioned in the edited document:
- there are many carnivores around, how did you know it was only leopard conflict? How ensure attack could be attributed to leopards?
- You almost completely neglect to mention a primary issue with this kind of deterrent - efficacy over time and habitation of carnivores. You indicate the local guardians randomly turn on/off the fox lights but this needs more context. Also you need to mention this as a potential negative in the discussion.
- In terms of analysis, presence of fox lights certainly doesn't completely prevent attacks on livestock, but did it change the number of incidents or size of the incident? Looking at the raw data, seems like places where deterrents were had only 1 or 2 deaths (if any), whereas locations with no fox lights had 4, 5 or more deaths. Was that a result of more attacks, or that a single attack resulted in more deaths?

---

## Round 0.2 · Minor Revisions

One of my reviewers has not seen the need to add any more comments, the other had gone to some trouble to make detailed edits. You should make those changes quickly and get the revision back to me, when I expect to accept the paper.

·

Basic reporting

Clear professional English. Definitely improved since first draft.

Experimental design

Improved as well.

Validity of the findings

Substantially improved. The conclusions are now much better tied to the results of the study.

Additional comments

Title – recommended change to ‘of the Himalayan Mountains’ or Himalayan region...
L12 – recommended addition ‘with trophic cascades on ecosystems and questionable impacts on human-wildlife conflict.’ Would be good to double-check some studies on this topic.
L24 – what were the ‘community-based conservation initiatives’ that you mention? Seems like you implemented fox lights but that was it? I’m not sure that falls under ‘community-based conservation.’
L25 – this is a ‘large semi-natural landscape’ I think

Intro
L44-46 – who are ‘marginal’ livestock owners? Are they not very good? They have small herds? This sentence and line of reasoning needs better explanation and possibly a citation.
L53 – add comma after hence
L56-58 – as in the Abstract, you may also want to state the questionable effectiveness of the lethal measures as an additional deterrent from that method and positive of non-lethal methods.
L62 – audio or visual deterrents, physical barriers etc.
L67 – don’t forget the striped hyena!
L93 – I was under the impression that its not just older less-fit or sick/injured males but also young adolescent or dispersing males that are also conflict-prone
L118 – make this last sentence the first sentence of your next paragraph. It is the key reason you did this work! Don’t hide it.
L126-130 – clarify when fox lights vs lion lights have been tested in the various studies.
L132 – the meaning of the last sentence is unclear

Methods
L170 – it is unclear, did you simply inform people about fox lights or did you tell them about the importance of carnivores, their plight, the variety of non-lethal intervention measures and, importantly, instruct and/or train people to implement these measures? Was the only thing implemented by the villagers actually the fox lights or were other measures implemented?
L225 – I’m curious why the presence or absence of water (or distance to water/stream), and topographic complexity (aka topographic roughness or similar measures) were not considered?
L239 – Was # of households and/or # of people also considered here? How good a correlation is there between human presence and night-lights in this part of India?
L242 – Explain this concept better. Altitude would not necessarily impact ecology of predators, or higher elevations cause them to predate more on livestock. What do you think is behind this? Also, check citations again – Rostro-Garcia find it peaks between 1000 and 3000m and then goes back down. And Miller is just a review article – better to cite the original articles (which both happen to be from western Hemisphere…) Personally I’d expect topographic ruggedness to be more important than mean elevation in most cases…

L312 – in discussion then, a suggestion would be installtion of lights that help prevent depredation at night and (presumably) better herding practices during the day could really minimize overall depredation events.
L315 – in this section you specifically mention some characteristics but do not mention variation in the vegetation characteristics (like % tree cover or dense forest etc.). Those could be reported too.

Discussion
L351 – it is ‘evidence-based results’ and ‘human-dominated landscapes’
L356 – add in the fact that there was significant decline in livestock depredation by leopard but no difference in leopard visitation or presence between the sites.
L359 – you found no support for hypotheses such as veg cover or open habitats on ‘livestock predation’ not on ‘efficacy of fox lights.’ Correct? Of the 3 hypotheses you list none mention any impact of socio-economic or landscape scale variables on the efficacy of fox lights.
L362 – livestock depredation was random in nature with respect to the variables you measured
L366 – again I’m unclear the link between your experiment with installing fox lights and link to ‘community-based conservation programs.’ To me they seem like different things entirely. You tested the efficacy of fox lights on reducing leopard depredation but I don’t see any connection to community-based conservation programs that do things like educational scholarships, awareness-raising, agriculture or livestock husbandry trainings etc. If there was stuff like that, then I’d expect those type of variables (i.e. adherence to improved livestock herding or building better corrals etc.) to have been measured and included in the models.
L369 – paragraph starting on L369 needs a topic sentence and better link from previous paragraph
L399 – we found no evidence of animal practices impacting predation events by leopards…
L409 – one of the best studies here is Lichtenfeld et al. 2015 ‘evidence-based…’
L416 - community-based conservation; community-based conflict approach (L420). Review for other instances where dashes are needed throughout the MS.
L419 – this is a semi-natural ecosystem
L428 – this paragraph sounds more like a concluding paragraph to me. Consider moving it down to last paragraph before Conclusions section.
L445 – Third, we couldn’t measure…
L455 – what is meant by ‘latitudinal and longitudinal’? Do you just mean studies should be conducted under varying conditions globally?
L460 – remove extra comma.
L460 – how is this work a community-based model? Our work can be successfully replicated…
L463 – ‘due to variation in animal behavior and environmental or social factors.’
L463 – ‘The majority of predator…’
L464 – ‘long-term’
L465 – hence,

---

## Round 0.3 · accepted · Accept

Thank you for addressing the reviewer's comments. I found your responses took care of the remaining issues.